# Dual Advancement of Representation Learning and Clustering for Sparse and Noisy Images

## ABSTRACT

Sparse and noisy images (SNIs), like those in spatial gene expression data, pose significant challenges for effective representation learning and clustering, which are essential for thorough data analysis and interpretation. In response to these challenges, we propose **D**ual **A**dvancement of **R**epresentation **L**earning and **C**lustering (**DARLC**), an innovative framework that leverages contrastive learning to enhance the representations derived from masked image modeling. Simultaneously, *DARLC* integrates cluster assignments in a cohesive, end-to-end approach. This integrated clustering strategy addresses the "class collision problem" inherent in contrastive learning, thus improving the quality of the resulting representations. To generate more plausible positive views for contrastive learning, we employ a graph attention network-based technique that produces denoised images as augmented data. As such, our framework offers a comprehensive approach that improves the learning of representations by enhancing their local perceptibility, distinctiveness, and the understanding of relational semantics. Furthermore, we utilize a Student's t mixture model to achieve more robust and adaptable clustering of SNIs. Extensive evaluation on 12 real-world datasets of SNIs, representing spatial gene expressions, demonstrat *DARLC*'s superiority over current state-of-the-art methods in both image clustering and generating representations that accurately reflect biosemantics content and gene interactions.

## CCS CONCEPTS

• **Computing methodologies** → **Computer vision**.

## KEYWORDS

Representation Learning, Clustering

## 1 INTRODUCTION

Sparse and noisy images (SNIs), commonly encountered in specialized fields like biomedical sciences, astronomy, and microscopy [33, 39, 48], are characterized by extensive uninformative regions (e.g., voids or background areas), considerable image noise, and severely fragmented visual patterns. These characteristics significantly increase the complexity in analysis and interpretation. A prime example is spatial gene expression Pattern (SGEP) images generated through spatial transcriptomics (ST) technology [35]. As illustrated in Figure 1, the high levels of sparsity and noise of an SGEP image

Unpublished working draft. Not for distribution.

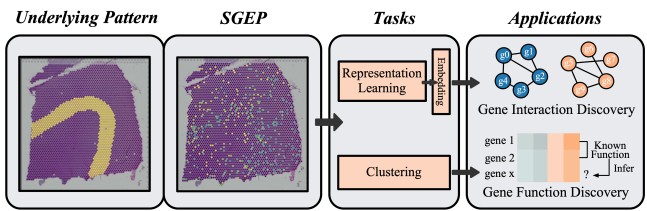

**Figure 1: An example of sparse and noisy SGEP image is displayed in the second panel to the left. The gene expression levels across space are represented by pixel brightness, while void expression areas are displayed in purple. The first panel showcases the regions (cortical layer V) where the gene is significantly expressed, namely the major gene expression pattern. The right two panels illustrate the tasks and applications that utilize SGEP images.**

complicate the discerning of its underlying major gene expression pattern.

Image clustering can group unlabelled images into distinct clusters, facilitating the exploration of image-implied semantics or functions. For example, clustering SGEP images offers a cost-effective means to identify groups of cofunctional genes and infer gene functions [34]. To obtain informative clustering results, it is essential to learn meaningful image representations, for which self-supervised learning (SSL) is the predominant approach in general scenarios. These include contrastive learning (CL) methods, exemplified by MoCo [20], and masked image modeling (MIM) such as MAE [19]. These methods offer distinct learning perspectives: MIM methods tend to learn local context-aware, holistic features for reconstruction tasks [19], while CL methods focus on learning instance-wise discriminative features [21]. Acknowledging the synergistic advantages of these methodologies, some researchers are endeavoring to utilize CL to refine the representations acquired through MIM [21, 47]. Moreover, many efforts [25, 32, 41] are directed towards guiding the process of learning representations through clustering tasks. This is achieved by jointly learning representations and executing clustering in an integrated and end-to-end manner, yielding image representations that are well-suited for clustering tasks.

However, for SNIs, both representation learning and clustering present significant challenges. Firstly, the widespread presence of uninformative voids or background areas, along with elevated noise levels and extremely fragmented visual patterns, substantially impedes the extraction of semantically meaningful visual features. This challenge has been highlighted in prior studies [30] and is further corroborated by our experiments (see Supplementary Table 1). Secondly, the inherent random noise across pixels induces considerable variability in visual patterns, even among images of

the same category [34], exposing clustering algorithms to a high overfitting risk [1].

Inspired by the aforementioned works, in order to better analyze SNIs, we propose a novel and unified framework, named **J**oint **L**earning of **R**epresentations and **C**luster **A**ssignments (*DARLC*). This framework not only leverages CL to boost the representation learned by MIM but also jointly learns cluster assignments in a self-paced and end-to-end manner, further refining the representation. Nonetheless, our experiments (see Supplementary Table 2) showcase that conventional data augmentation techniques (e.g., cropping and rotating) are ineffective for SNIs, as the augmented images often contain substantial void regions and noise, hampering the extraction of informative visual features. To overcome this limitation, we introduce a data augmentation method based on a graph attention network (GAT) [13, 37] that aggregates information from neighboring pixels to enhance visual patterns, generating smoothed images that act as more plausible positive views so as to improve the effectiveness of contrastive learning.

Additionally, we observed that the clustering algorithms used in many deep clustering methods are either sensitive to outliers, as demonstrated by the Gaussian mixture model (GMM) in DAGMM [49] and manifold clustering in EDESC [3], or lack the flexibility to different data distributions, such as the inflexible Cauchy kernel-based method in DEC [41]. In response, *DARLC* employs a specialized nonlinear projection head to normalize image embeddings, aligning them more closely with a t-distribution. This is followed by modeling with a Student's t mixture model (SMM) for soft clustering. SMM provides a more robust solution by down-weighing extreme values and is more adaptable by altering the degrees of freedom, making it particularly suitable for clustering in the context of SNIs. Furthermore, the clustering loss in *DARLC* also serves to regularize CL, alleviating the "class collision problem" that stems from false negative pairs in CL [4]. Unlike existing regularized CL methods [12, 24, 45], which directly integrate clustering into the CL throughout training, this clustering follows the "warm-up" representation learning, significantly expediting training convergence and enhancing clustering accuracy, as demonstrated in our ablation study. All these features collectively contribute to the finding that *DARLC*-generated image representations not only enhance clustering performance but also exhibit improvements in other semantic distance-based tasks, such as the discovery of functionally interactive genes. In summary, our main contributions are:

- We propose *DARLC*, a novel unified framework for joint representation learning and clustering of SNIs. *DARLC* marks the first endeavor in integrating contrastive learning, MIM and deep clustering into a cohesive process for representation learning. The resultant representations not only enhance image clustering performance but also benefit other semantic distance-based tasks.
- *DARLC* has developed a data augmentation method more suitable for SNIs, using a GAT to generate smoothed images as plausible positive views for CL.
- An SMM-based method is designed to cluster SNIs in a more robust and adaptable manner. Additional features of this clustering method include a novel Laplacian loss for guiding the initial phase of clustering, and a differentiable cross-entropy hinge loss

for controlling cluster sizes. This clustering also addresses the class collision problem by pulling close related instances.

- Extensive experiments have been conducted across 12 real SGEP datasets [28]. Our results show that *DARLC* surpasses the state-of-the-art (SOTA) methods in both image clustering and generating image representations that accurately capture gene interactions.

## 2 RELATED WORKS

### 2.1 Self-supervised Representation Learning for Images

Most related SSL studies include CL and MIM methods. In CL, an input instance forms positive pairs with its augmented views, while forms negative pairs with other instances. Paradigmatic CL methods aim to learn instance discriminative representations by maximizing the similarity between positive pairs while minimizing it between negative pairs in a latent space [8, 20]. To address the class collision problem due to false negative samples, several studies [12, 24, 45] regularize CL with clustering, while others [5, 6, 9, 17] bypass the using of negative samples altogether. In contrast, MIM methods focus on learning local context-aware features by restoring raw pixel values from masked image patches [2, 16, 19, 42]. Several researchers are realizing the advantages of integrating these methodologies and endeavoring to utilize CL to refine MIM-generated representations [21, 42]. For instance, iBOT [47] contrasts between the reconstructed tokens of masked and unmasked image patches. Yet, to the best of our knowledge, *DARLC* is the first method that learns image representations from all aspects of discriminability, local perceptability, and relational semantic structures.

### 2.2 Deep Image Clustering.

Related deep image clustering studies include deep autoencoder-based methods [32], which couple representation learning with deep embedded clustering in an end-to-end manner, as exemplified by methods like DEC [18, 23, 41]. Subsequent improvements to DEC focus on strategies like overweighing reliable samples (e.g., IDCEC [29] ), and replacing Euclidean distance-based clustering with deep subspace clustering (e.g., EDESC [3]) or GMM-based clustering [38, 49]. Recent studies, including CC [25], DCP [27], CVCL [7], and CCES [43], directly integrate CL into the clustering process by contrasting at both instance and cluster levels across views, generating a soft cluster assignment matrix as deep embeddings for iterative refinement. Compared to these methods, *DARLC* offers a more comprehensive and potent mechanism for learning deep embeddings, a more robust and adaptable clustering algorithm, and a warm-up representation learning phase for accelerating clustering convergence.

## 3 METHODOLOGY

### 3.1 Overview

The framework of *DARLC*, as illustrated in Figure 2 and Algorithm 1, comprises two modules: a self-supervised representation learning and a deep clustering. The self-supervised representation learning module unifies CL and MIM, encompassing three encoders: an online encoder and a target momentum encoder for the contrastive

## Module I: Representation Learning | Module II: Cluster Assignment

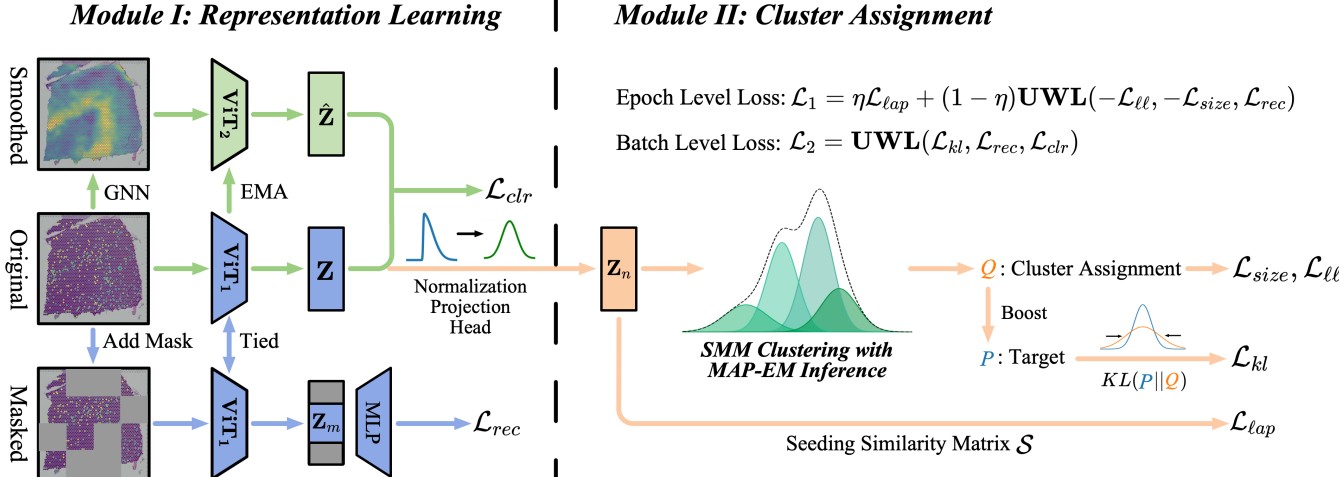

**Figure 2: The framework of *DARLC* consists of two components: the representation learning and the deep clustering. The representation learning component integrates MIM and CL to generate image embeddings, which are then normalized through a non-linear projection head. Normalized representations are modeled by an SMM to derive their soft cluster assignments, which are used to construct various loss functions. With these loss functions, the two components are jointly optimized in a self-paced and end-to-end manner.**

branch, and a masked encoder for the MIM branch. All three encoders adopt the identical vision transformer (ViT) [14] architecture, with shared parameters between the online encoder and masked encoder. The parameters of the target momentum encoder are updated using exponential moving average (EMA), as suggested by BYOL [17]. The initial phase of the unified representation learning involves a warm-up pretraining to generate preliminary embeddings, which are then normalized by a nonlinear projection head. This normalization aligns the embeddings more closely with t-distributions, setting the stage for subsequent t-distribution-based clustering. The deep clustering module utilizes a SMM to cluster the normalized embeddings, generating soft cluster assignment scores involved in the calculation of various loss functions. There are two types of loss functions: $\mathcal{L}_1$, an epoch-level loss function for maximizing the empirical likelihood of observed instances, and $\mathcal{L}_2$, a batch-level loss function for discriminatively boosted clustering optimization [41]. The two loss functions work in tandem, enabling the joint refinement of image embeddings and cluster assignments in a self-paced and end-to-end manner.

### 3.2 Unified Self-supervised Representation Learning (Module I)

*3.2.1 Denoising-based Data Augmentation.* We train a graph attention autoencoder $\mathcal{G}$ to generate smoothed images, serving as augmented positive instances [13, 37]. Initially, for each image, we construct an undirected and unweighted graph by treating pixels as nodes connected to their k-nearest neighbors. Specifically, for a given image $\mathbf{X} \in \mathbb{R}^{C \times H \times W}$, where $C$ is the number of channels, $H$ and $W$ are the height and width of the image, respectively. Let $N_{pix} = H \times W$ denote the number of pixels, $\mathbf{v}_\iota \in \mathbb{R}^C$ denote the pixel vector at location $\iota$, $\forall \iota \in \{1, 2, ..., N_{pix}\}$. The encoder in $\mathcal{G}$ comprises $L$ layers. For each layer $t \in \{1, 2, ...L - 1\}$, with the initial

value $\mathbf{h}_\iota^{(0)} = \mathbf{v}_\iota$, the output $\mathbf{h}_\iota^{(t)} \in \mathbb{R}^{d_p}$ is calculated as follows:

$$\mathbf{h}_\iota^{(t)} = LeakyReLU\left(\sum_{\nu \in S_\iota} \mathbf{att}_{\iota\nu}(\mathbf{W}^{(t)}\mathbf{h}_\nu^{(t-1)})\right), \quad (1)$$

where $\mathbf{W}^{(t)}$ represents the trainable weights of the $t$-th autoencoder layer, $S_\iota$ the set of node $\iota$'s neighbors within a pre-specified radius $r$. The attention score, $\mathbf{att}_{\iota\nu}$, between nodes $\iota$ and $\nu$ are computed as follows:

$$\alpha_{\iota\nu}^{(t)} = \mathbf{w}_{att}^{(t)} LeakyReLU(\mathbf{W}^{(t)}[\mathbf{h}_\iota^{(t-1)}||\mathbf{h}_\nu^{(t-1)}]), \quad (2)$$

$$\mathbf{att}_{\iota\nu}^{(t)} = \frac{\exp(\alpha_{\iota\nu}^{(t)})}{\sum_{\iota \in S_\iota} \exp(\alpha_{\iota\nu}^{(t)})}, \quad (3)$$

where $\mathbf{w}_{att}^{(t)}$ represents the trainable attention weights. The decoder of $\mathcal{G}$ mirrors the encoder with tied weights. The total loss function is defined as:

$$L_{denoise} = \sum_{\iota=1}^{N_{pix}} \|\mathbf{v}_\iota - \widetilde{\mathbf{h}}_\iota^{(0)}\|_2, \quad (4)$$

where $\widetilde{\mathbf{h}}_\iota^{(0)}$ denotes the reconstructed $\mathbf{v}_\iota$ output by the decoder. Once trained, $\mathcal{G}$ is applied to any given image $\mathbf{X}_i$, to generate a smoothed image $\overline{\mathbf{X}}_i$ for a given image $\mathbf{X}_i$ as:

$$\overline{\mathbf{X}}_i = \mathcal{G}(\mathbf{X}_i, \mathbf{W}, w_{att}) \quad (5)$$

*3.2.2 Unified Self-supervised Image Representation Learning.* This SSL model encompasses two branches: a MIM branch $\mathcal{M}$ and a contrastive branch $C$. $\mathcal{M}$ is an adapted version of MAE, specifically designed for generating image patch embeddings in the context of SNIs. In this adaption, the standard MAE encoder is replaced with a lightweight ViT encoder, denoted as $\mathcal{M}_E$, with four transformer blocks, four attention heads, and a higher masking ratio (80%). Meanwhile, the original transformer-based MAE decoder,

**Algorithm 1** Algorithm for Dual Advancement of Representations Learning and Clustering (*DARLC*).

---

**Input:** Images $\mathcal{X} \in \mathbb{R}^{N \times C \times H \times W}$; Seeding similarity matrix $\mathcal{S} \in \mathbb{R}^{N \times N}$; Denoise GAT $\mathcal{G}$; Maximum epochs $E_{max}$; Number of images $N$; Number of clusters $K$.

**Definition:** Parametes of $\cdot$ are denoted as $\Pi(\cdot)$; MIM branch $\mathcal{M}$; CL branch $C$; Projection head $\mathcal{H}$; **UWL** (Uncertain Weight Loss function); **EMA** (Exponential Moving Average function).

**Output:** Representations $Z \in \mathbb{R}^{N \times D}$; Soft clustering $Q \in \mathbb{R}^{N \times K}$.

1: Compute smoothed image $\bar{\mathcal{X}}$ by Eq. (5).
2: **while** $epoch < E_{max}$ **do**
3:     **for** $\mathbf{X}_b, \overline{\mathbf{X}}_b$ in $\mathcal{X}, \bar{\mathcal{X}}$ **do**
4:         Compute $\widehat{\mathbf{X}}_b$ by Eq. (6) $\rightarrow \mathcal{L}_{rec}(\mathbf{X}_b, \widehat{\mathbf{X}}_b)$ by Eq. (7).
5:         Compute $\mathbf{e}_b, \overline{\mathbf{e}}_b$ by Eq. (9) $\rightarrow \mathcal{L}_{clr}(\mathbf{e}_b, \overline{\mathbf{e}}_b)$ by Eq. (10).
6:         $\mathcal{L}_{ssl} = \mathbf{UWL}(\mathcal{L}_{rec}, \mathcal{L}_{clr})$.
7:         Update $\Pi(\mathcal{M})$ using $\mathcal{L}_{ssl}$.
8:         Update $\Pi(C)$ with $\mathbf{EMA}(\Pi(\mathcal{M}))$.
9:     **end for**
10: **end while**
11: **while** (not converged) & ($epoch < E_{max}$) **do**
12:     Compute $\widehat{\mathcal{X}}$ by Eq. (6) $\rightarrow \mathcal{L}_{rec}(\mathcal{X}, \widehat{\mathcal{X}})$ by Eq. (7).
13:     Compute $\mathbf{Z}$ by Eqs. (9), (12) $\rightarrow \mathcal{L}_{\ell ap}(\mathbf{Z}, \mathcal{S})$ by Eq. (19).
14:     SMM parameters inference using MAP-EM $\rightarrow \Theta$.
15:     $Q = \text{SMM}(\mathbf{Z}|\Theta) \rightarrow \mathcal{L}_{size}(Q), \mathcal{L}_{\ell\ell}(Q)$ by Eqs. (22), (20).
16:     $\mathcal{L}_1 = \eta \mathcal{L}_{\ell ap} + (1 - \eta)\mathbf{UWL}(-\mathcal{L}_{\ell\ell}, -\mathcal{L}_{size}, \mathcal{L}_{rec})$.
17:     Update $\Pi(\mathcal{M}), \Pi(C), \Pi(\mathcal{H})$ using $\mathcal{L}_1$.
18:     **for** $\mathbf{X}_b, \overline{\mathbf{X}}_b$ in $\mathcal{X}, \overline{\mathcal{X}}$ **do**
19:         Compute $P$ by Eq. (24) $\rightarrow \mathcal{L}_{kl}(P, Q)$ by Eq. (23).
20:         Compute $\widehat{\mathbf{X}}_b$ by Eq. (6) $\rightarrow \mathcal{L}_{rec}(\mathbf{X}_b, \widehat{\mathbf{X}}_b)$ by Eq. (7).
21:         Compute $\mathbf{e}_b, \overline{\mathbf{e}}_b$ by Eq. (9) $\rightarrow \mathcal{L}_{clr}(\mathbf{e}_b, \overline{\mathbf{e}}_b)$ by Eq. (10).
22:         $\mathcal{L}_2 = \mathbf{UWL}(\mathcal{L}_{kl}, \mathcal{L}_{rec}, \mathcal{L}_{clr})$.
23:         Update $\Theta$ and $\Pi(\mathcal{M}), \Pi(C), \Pi(\mathcal{H})$ using $\mathcal{L}_2$.
24:     **end for**
25: **end while**
26: **return** $Z, Q$

---

is substituted with a fully-connected linear decoder $\mathcal{M}_D$. For any given image $\mathbf{X}_i$, the regenrated image $\widehat{\mathbf{X}}_i$ is as follows:

$$\widehat{\mathbf{X}}_i = \mathcal{M}_D(\mathcal{M}_E(\mathbf{X}_i, \mathbf{W}_E), \mathbf{W}_D) \tag{6}$$

The MIM branch loss for the current batch are:

$$\mathcal{L}_{rec} = \frac{1}{N_b} \sum_{i=1}^{N_b} \frac{1}{N_{masked}} \sum_{j \in S_{masked}} (\mathbf{p}_{i,j} - \widehat{\mathbf{p}}_{i,j})^T (\mathbf{p}_{i,j} - \widehat{\mathbf{p}}_{i,j}), \tag{7}$$

where $N_b$ is the batch size, $\mathbf{W}_E$ and $\mathbf{W}_D$ represent the parameters of the encoder and decoder, respectively. $N_{masked}$ denotes the number of masked patches, $S_{masked}$ the set of masked patches. $\mathbf{p}_{i,j}$ and $\widehat{\mathbf{p}}_{i,j}$ represent the original and regenerated $j$-th image patch of $\mathbf{X}_i$, respectively.

For the contrastive branch $C$, the $N_b$ raw images form positive pairs with their respective smoothed images, and negative pairs with the other $2N_b - 2$ images in the same batch. $C$ is structured around a pseudo-siamese network with two encoders: an online encoder $C_O$ and a target momentum encoder $C_T$. Both encoders

share the identical network architecture as $\mathcal{P}_E$, with $C_O$ and $\mathcal{P}_E$ having tied parameters. The parameters of $C_T$ are updated using EMA. Concretely, let $\widetilde{\mathbf{W}}_E$ and $\overline{\mathbf{W}}_E$ denote the parameters of the $C_O$ and $C_T$, respectively. Then we have:

$$\widetilde{\mathbf{W}}_E = \mathbf{W}_E, \quad \overline{\mathbf{W}}_E = m\overline{\mathbf{W}}_E + (1 - m)\widetilde{\mathbf{W}}_E \tag{8}$$

Here, $m$ represents the momentum, fixed at 0.999. For each image $\mathbf{X}_i$ and its augmented counterpart $\overline{\mathbf{X}}_i$, their respective embedding vectors, $\mathbf{e}_i$ and $\overline{\mathbf{e}}_i \in \mathbb{R}^D$, are obtained as: $\mathbf{e}_i = \tilde{g}(C_O(\mathbf{X}_i, \widetilde{\mathbf{W}}_E))$, $\overline{\mathbf{e}}_i = \bar{g}(C_T(\overline{\mathbf{X}}_i, \overline{\mathbf{W}}_E))$,

$$\mathbf{e}_i = \tilde{g}(C_O(\mathbf{X}_i, \widetilde{\mathbf{W}}_E)), \quad \overline{\mathbf{e}}_i = \bar{g}(C_T(\overline{\mathbf{X}}_i, \overline{\mathbf{W}}_E)), \tag{9}$$

where $\tilde{g}$ and $\bar{g}$ are linear mapping functions with trainable weights, and the weights of $\bar{g}$ are updated using EMA as well. The contrastive loss $\mathcal{L}_{clr,i}$ is computed as :

$$\mathcal{L}_{clr,i} = -\log \frac{\mathbf{s}(\mathbf{e}_i, \overline{\mathbf{e}}_i)}{\sum_{k=1, k \neq i}^{N_b} \mathbf{s}(\mathbf{e}_i, \mathbf{e}_k) + \sum_{k=1}^{N_b} \mathbf{s}(\mathbf{e}_i, \overline{\mathbf{e}}_k)}$$
$$- \log \frac{\mathbf{s}(\overline{\mathbf{e}}_i, \mathbf{e}_i)}{\sum_{k=1}^{N_b} \mathbf{s}(\overline{\mathbf{e}}_i, \mathbf{e}_k) + \sum_{k=1, k \neq i}^{N_b} \mathbf{s}(\overline{\mathbf{e}}_i, \overline{\mathbf{e}}_k)}, \tag{10}$$

where $\mathbf{s}(\cdot, \cdot) = \exp(\cos(\cdot, \cdot)/\tau)$, and $\tau$ is a temperature coefficient, defaulting to 0.5. Consequently, the loss function $\mathcal{L}_{clr}$ is defined as $\mathcal{L}_{clr} = \frac{1}{N_b} \sum_{i=1}^{N_b} \mathcal{L}_{clr,i}$. Finally, $\mathcal{L}_{rec}$ and $\mathcal{L}_{clr}$ are dynamically integrated into the total SSL loss function $\mathcal{L}_{ssl}$ using the uncertain weights loss (**UWL**) function [26]:

$$\mathcal{L}_{ssl} = \mathbf{UWL}(\mathcal{L}_{rec}, \mathcal{L}_{clr})$$
$$= \frac{1}{2\sigma_1^2} \mathcal{L}_{rec} + \frac{1}{2\sigma_2^2} \mathcal{L}_{clr} + \log(1 + \sigma_1^2) + \log(1 + \sigma_2^2), \tag{11}$$

where $\sigma_1$ and $\sigma_2$ are trainable noise parameters.

## 3.3 Self-paced Deep Image Clustering (Module II)

*3.3.1 Student's t mixture model.* Let $\mathbf{e}_i$ denote the *Module I*-generated embedding vector for the $i$-th original image. We first map $\mathbf{e}_i$ to $\mathbf{z}_i \in \mathbb{R}^D$ in a latent space wherein it is more conformed to a t-distribution. This mapping is achieved through a nonlinear projection head with batch normalization and scaled exponential linear unit (SELU) activation function:

$$\mathbf{z}_i = SELU(BN(\mathbf{W}_p, \mathbf{e}_i)), \tag{12}$$

The set of these vectors, $\mathbf{Z} = \{\mathbf{z}_i\}_{i=1}^N \in \mathbb{R}^{N \times D}$, is modeled using an SMM, whose components correspond to image clusters. Since extreme values are downweighed by SMM during parameter inference, this clustering is more robust to outliers and variances. The SMM is parameterized by $\Theta = \{\theta_k : \pi_k, \mu_k, \Sigma_k, v_k, \forall k \in K\}$, where $K$ represents the number of components and is assumed to be known or can be automatically inferred (see Section 4.1). Here, $\pi_k, \mu_k, \Sigma_k, v_k$ denote the weight, mean, covariance matrix, and degree of freedom of the $k$-th component, respectively. The density function of $\mathbf{z}_i$ is expressed as:

$$p(\mathbf{z}_i|\Theta) = \sum_{k=1}^K \pi_k \, \phi(\mathbf{z}_i|\mu_k, \Sigma_k, v_k) \tag{13}$$

For robust model inference, we use the maximum a posterior-expectation maximization (MAP-EM) algorithm by applying a conjugate Dirichlet prior on $\Pi = \{\pi_k, \forall k \in [1, K]\}$ and a normal inverse Wishart (NIW) prior on $\mu_k, \Sigma_k$:

$$\Pi \sim Dir(\Pi|\alpha^0), \tag{14}$$

$$\mu_k, \Sigma_k \sim NIW(\mu_k, \Sigma_k|\mathrm{m}_0, \kappa_0, S_0, \rho_0), \forall k \in [1, K] \tag{15}$$

To simplify the inference, we rewrite the Student's t density function $\phi$ as a Gaussian scale mixture by introducing an "artificial" hidden variable $\zeta_{i,k}$, $\forall i \in [1, N]$, $\forall k \in [1, K]$ that follows a Gamma distribution parameterized by $v_k$:

$$\phi(\mathbf{z}_i|\mu_k, \Sigma_k, v_k) =$$
$$\int \mathcal{N}\left(\mathbf{z}_i\middle|\mu_k, \frac{\Sigma_k}{\zeta_{i,k}}\right) \Gamma\left(\zeta_{i,k}\middle|\frac{v_k}{2}, \frac{v_k}{2}\right) d\zeta_{i,k} \tag{16}$$

We further introduce a missing variable $\xi_i$ to represent the component membership of $\mathbf{z}_i$. The posterior complete data log likelihood is then expressed as:

$$\ell_c(\Theta) = \log P(\mathbf{Z}, \zeta, \xi|\Theta)$$
$$= \log Dir(\Pi|\alpha^0) +$$
$$\sum_k \log NIW(\mu_k, \Sigma_k|\mathrm{m}_0, \kappa_0, S_0, \rho_0) +$$
$$\sum_i \sum_k [II(\xi_i = k)(\log \pi_k + \log \Phi(\mathbf{z}_i, \zeta_{i,k}|\mu_k, \Sigma_k, v_k))] \tag{17}$$

In the $t$-th iteration of the E-step, the expected sufficient statistics $\overline{\xi_{i,k}}^{(t)}$ and $\overline{\zeta_{i,k}}^{(t)}$ are derived based on $\Theta^{(t-1)}$. In the subsequent M-step, $\Theta^{(t-1)}$ is updated to $\Theta^{(t)}$ by maximizing the auxiliary function $Q(\Theta, \Theta^{(t-1)}) = E(\ell_c(\Theta)|\Theta^{(t-1)})$. These two steps are alternated until either convergence is reached or a predefined maximum number of iterations is attained. Refer to Supplementary 1.1 for details of the model inference.

### 3.3.2 Self-paced Joint Optimization of Image Embeddings and Cluster Assignments.
Two loss functions, $\mathcal{L}_1$ and $\mathcal{L}_2$, are calculated based on clustering results for updating parameters of both *Module I* and the SMM through loss gradient backpropagation. This iterative process progressively improves the clustering-oriented image embeddings and clustering results. Upon completing the inference of SMM parameters $\widetilde{\Theta}$ in each epoch, an epoch-level loss $\mathcal{L}_1$ is calculated for updating parameters of *Module I*:

$$\mathcal{L}_1 = \eta \mathcal{L}_{\ell ap} + (1 - \eta)\mathbf{UWL}(-\mathcal{L}_{\ell\ell}, -\mathcal{L}_{size}, \mathcal{L}_{rec}). \tag{18}$$

Here, $\mathcal{L}_{\ell ap}$ is a Laplacian regularization term that promotes the similarities among image embeddings $\mathbf{Z}$ to be consistent with a seeding image-image similarity matrix $\mathcal{S}$, informing the initial training phase. The derivation of $\mathcal{S}$ is detailed in Supplementary 1.2. $\mathcal{L}_{\ell ap}$ is defined as follows:

$$\mathcal{L}_{\ell ap} = Tr\left(Z^T \left(I - \mathcal{D}^{-\frac{1}{2}} \mathcal{S} \mathcal{D}^{-\frac{1}{2}}\right) Z\right), \tag{19}$$

where $\mathcal{D}$ is the degree matrix of $\mathcal{S}$, and $\eta$, initially set at 0.5, decays over the training course so that the influence of $\mathcal{S}$ is gradually reduced. $\mathcal{L}_{\ell\ell}$ represents the log likelihood of the embeddings given the estimated SMM parameters $\widetilde{\Theta}$:

$$\mathcal{L}_{\ell\ell} = \sum_{i=1}^{N} \log\left[\sum_k q_{i,k}\right], \tag{20}$$

$$q_{i,k} = \pi_k \phi(\mathbf{z}_i|\mu_k, \Sigma_k, v_k), \forall i \in [1, N], \forall k \in [1, K]. \tag{21}$$

$\mathcal{L}_{size}$ penalizes empty and tiny clusters, while exempting those whose size exceeds a predefined threshold $v$ so that image assignments is not overly uniform:

$$\mathcal{L}_{size} = \sum_{k=1}^{K} -J_k \log J_k, J_k = \begin{cases} \frac{\sum_i^N q_{i,k}}{N} & , if J_k \leq v \\ 1 & . otherwise \end{cases} \tag{22}$$

$\mathcal{L}_{rec}$, defined in Equation 7, aims to enhance the local-context awareness of embeddings. Subsequently, within the same epoch, a batch-level loss $\mathcal{L}_2 = \mathbf{UWL}(\mathcal{L}_{kl}, \mathcal{L}_{rec}, \mathcal{L}_{clr})$ is utilized to update *Module I* and SMM parameters across successive batches. Here, $\mathcal{L}_{rec}$ and $\mathcal{L}_{clr}$ remains same as in Equations 7 and **??** except being calculated on the batch-level. $\mathcal{L}_{kl}$ boosts high-confidence images, incrementally grouping similar instances while separating dissimilar ones:

$$\mathcal{L}_{kl} = KL(\mathcal{P}|Q) = \sum_i^N \sum_j^K \mathbb{p}_{i,j} \log \frac{\mathbb{p}_{i,j}}{\mathbb{q}_{i,j}}, \tag{23}$$

$$\text{where } \mathbb{q}_{i,k} = \frac{q_{i,k}}{\sum_c q_{i,c}}, \mathbb{p}_{i,k} = \frac{\mathbb{q}_{i,k}^2/\sum_i \mathbb{q}_{i,c}}{\sum_c \left(\mathbb{q}_{i,c}^2/\sum_i \mathbb{q}_{i,c}\right)} \tag{24}$$

Here, $q_{i,k}$ is same as in Equation 21, $\mathbb{q}_{i,k}$ represents the probability of assigning $i$-th image to the $k$-th SMM component, and $\mathbb{p}_{i,k}$ an auxiliary target distribution that boosts up high-confidence images. After this joint optimization, the training progresses to the next epoch, iterating until the end of the training process. The mathematical derivations of gradients of $\mathcal{L}_1$ and $\mathcal{L}_2$ with respect to $\mathbf{W}_E$, $\mathbf{W}_D$ and $\Theta$ are detailed in Supplementary 1.3.

## 4 EXPERIMENTS

### 4.1 Experimental Settings

#### 4.1.1 Datasets.
To comprehensively evaluate *DARLC*, we utilize 12 real ST datasets derived from different tissue slices of the human dorsolateral prefrontal cortex (hDLPFC). These datasets display spatial expression levels of the whole genome across the hDLPFC tissue (approximately 18700 75x75 SGEP images per dataset). These datasets are named by tissue slice numbers (151507-151510, 151669-151676), and accessible through the spatialLIBD package [31] at http://spatial.libd.org/spatialLIBD.

#### 4.1.2 Data quality control and preprocessing.
We conform to the conventional procedure for preprocessing ST data, as implemented in the SCANPY package [40]. Specifically, we first remove mitochondrial and External RNA Controls Consortium spike-in genes. Then, genes detected in fewer than 10 spots are excluded. To preserve the spatial data integrity, we do not perform quality control on spatial spots. Finally, the gene expression counts are normalized by library size, followed by log-transformation.

**Table 1: Clustering performance of *DARLC* and the benchmark methods across 12 datasets of spatial gene expression images are quantified using DBIE and DBIP scores. Lower scores indicate better performance ,the best score for each dataset is bolded, and the second-best score is underlined. The score standard deviation is subscripted.**

| Method | DBIE↓ | | | | | | | | | | | |
|---|---|---|---|---|---|---|---|---|---|---|---|---|
| | 151507 | 151508 | 151509 | 151510 | 151669 | 151670 | 151671 | 151672 | 151673 | 151674 | 151675 | 151676 |
| DEC | $10.88_{0.11}$ | $\underline{10.59}_{0.31}$ | $11.54_{0.14}$ | $10.88_{0.55}$ | $13.23_{1.87}$ | $11.15_{0.55}$ | $12.32_{0.97}$ | $11.05_{0.59}$ | $11.58_{1.20}$ | $11.22_{1.08}$ | $11.44_{0.70}$ | $11.42_{0.65}$ |
| DAGMM | $46.18_{27.97}$ | $47.42_{28.27}$ | $30.70_{18.39}$ | $19.02_{8.21}$ | $30.60_{21.35}$ | $24.81_{7.74}$ | $46.71_{42.62}$ | $54.80_{42.75}$ | $26.07_{5.72}$ | $39.47_{12.57}$ | $20.64_{17.48}$ | $42.83_{19.63}$ |
| EDESC | $12.55_{0.95}$ | $14.24_{1.39}$ | $11.81_{0.97}$ | $10.79_{0.88}$ | $12.48_{0.50}$ | $\underline{10.71}_{1.22}$ | $11.94_{1.08}$ | $13.53_{1.53}$ | $\underline{11.18}_{0.99}$ | $10.17_{1.70}$ | $\underline{9.94}_{1.44}$ | $\underline{10.04}_{1.12}$ |
| IDCEC | $15.65_{1.73}$ | $14.09_{0.88}$ | $13.55_{0.90}$ | $14.35_{1.31}$ | $\underline{11.18}_{0.13}$ | $11.85_{0.14}$ | $\underline{10.51}_{0.34}$ | $10.83_{0.49}$ | $11.23_{0.50}$ | $10.60_{0.34}$ | $12.30_{0.94}$ | $12.95_{1.10}$ |
| CC | $16.66_{0.02}$ | $18.66_{0.01}$ | $20.03_{0.04}$ | $16.38_{0.02}$ | $18.31_{0.04}$ | $18.74_{0.04}$ | $17.05_{0.03}$ | $19.05_{0.05}$ | $18.77_{0.04}$ | $19.53_{0.05}$ | $20.26_{0.02}$ | $17.41_{0.03}$ |
| DCP | $12.48_{0.47}$ | $12.07_{0.27}$ | $\underline{11.07}_{0.32}$ | $11.65_{0.24}$ | $11.45_{0.32}$ | $11.73_{0.10}$ | $11.61_{0.11}$ | $12.28_{0.39}$ | $11.92_{0.27}$ | $11.33_{0.86}$ | $12.09_{0.66}$ | $11.27_{0.03}$ |
| CVCL | $18.04_{7.64}$ | $33.14_{3.16}$ | $20.24_{4.30}$ | $28.77_{1.70}$ | $33.38_{4.85}$ | $55.02_{11.21}$ | $39.52_{11.33}$ | $31.00_{0.51}$ | $30.77_{1.70}$ | $31.15_{3.20}$ | $39.85_{7.30}$ | $30.96_{2.35}$ |
| iBOT-C | $\underline{10.75}_{0.01}$ | $10.88_{0.02}$ | $11.87_{0.01}$ | $\underline{10.23}_{0.03}$ | $13.15_{0.03}$ | $13.21_{0.02}$ | $12.69_{0.04}$ | $\underline{10.76}_{0.05}$ | $12.81_{0.02}$ | $\underline{10.11}_{0.01}$ | $10.71_{0.03}$ | $11.31_{0.01}$ |
| *DARLC* | $\mathbf{8.09}_{0.33}$ | $\mathbf{8.16}_{0.13}$ | $\mathbf{8.31}_{0.28}$ | $\mathbf{7.98}_{0.74}$ | $\mathbf{7.69}_{0.16}$ | $\mathbf{8.22}_{0.31}$ | $\mathbf{7.65}_{0.40}$ | $\mathbf{8.06}_{0.37}$ | $\mathbf{7.90}_{0.37}$ | $\mathbf{7.78}_{0.33}$ | $\mathbf{7.41}_{0.36}$ | $\mathbf{8.09}_{0.55}$ |

| Method | DBIP↓ | | | | | | | | | | | |
|---|---|---|---|---|---|---|---|---|---|---|---|---|
| | 151507 | 151508 | 151509 | 151510 | 151669 | 151670 | 151671 | 151672 | 151673 | 151674 | 151675 | 151676 |
| DEC | $\underline{2.38}_{0.06}$ | $\underline{2.18}_{0.02}$ | $2.68_{0.21}$ | $\underline{2.44}_{0.11}$ | $2.55_{0.10}$ | $\underline{2.49}_{0.09}$ | $3.04_{0.24}$ | $\underline{2.52}_{0.08}$ | $3.71_{0.53}$ | $3.32_{0.19}$ | $3.52_{0.59}$ | $\underline{2.42}_{0.08}$ |
| DAGMM | $16.19_{5.26}$ | $14.01_{4.23}$ | $12.35_{0.98}$ | $12.11_{2.09}$ | $15.65_{1.38}$ | $13.51_{3.34}$ | $31.03_{18.08}$ | $22.75_{12.10}$ | $27.17_{13.81}$ | $20.06_{2.27}$ | $16.14_{13.59}$ | $20.14_{9.84}$ |
| EDESC | $2.49_{0.17}$ | $2.66_{0.28}$ | $2.68_{0.32}$ | $2.57_{0.20}$ | $2.68_{0.36}$ | $2.81_{0.41}$ | $2.70_{0.36}$ | $2.66_{0.38}$ | $\underline{2.85}_{0.40}$ | $3.11_{0.14}$ | $\underline{2.75}_{0.30}$ | $2.77_{0.30}$ |
| IDCEC | $2.70_{0.05}$ | $2.52_{0.11}$ | $2.75_{0.07}$ | $2.66_{0.06}$ | $2.60_{0.28}$ | $2.65_{0.28}$ | $2.76_{0.26}$ | $2.69_{0.32}$ | $3.13_{0.23}$ | $2.99_{0.20}$ | $2.98_{0.26}$ | $3.07_{0.19}$ |
| CC | $3.18_{0.01}$ | $3.12_{0.01}$ | $4.28_{0.01}$ | $4.07_{0.03}$ | $3.13_{0.04}$ | $3.14_{0.01}$ | $4.12_{0.02}$ | $4.35_{0.01}$ | $3.18_{0.00}$ | $3.58_{0.00}$ | $2.93_{0.01}$ | $3.96_{0.04}$ |
| DCP | $2.46_{0.11}$ | $2.29_{0.03}$ | $\underline{2.54}_{0.12}$ | $2.46_{0.16}$ | $\underline{2.55}_{0.06}$ | $2.54_{0.10}$ | $\underline{2.59}_{0.06}$ | $2.55_{0.06}$ | $3.37_{0.03}$ | $\underline{2.96}_{0.16}$ | $3.00_{0.13}$ | $2.87_{0.16}$ |
| CVCL | $5.28_{1.35}$ | $5.68_{0.65}$ | $5.28_{0.70}$ | $5.72_{0.76}$ | $7.00_{2.07}$ | $8.26_{2.00}$ | $8.92_{2.02}$ | $6.69_{0.81}$ | $7.13_{0.84}$ | $5.45_{0.63}$ | $6.32_{1.07}$ | $4.97_{0.81}$ |
| iBOT-C | $3.39_{0.02}$ | $3.62_{0.04}$ | $4.77_{0.01}$ | $3.27_{0.03}$ | $4.90_{0.04}$ | $3.45_{0.02}$ | $3.48_{0.03}$ | $2.96_{0.02}$ | $4.87_{0.01}$ | $3.38_{0.01}$ | $3.57_{0.03}$ | $3.47_{0.02}$ |
| *DARLC* | $\mathbf{2.08}_{0.02}$ | $\mathbf{2.04}_{0.01}$ | $\mathbf{2.11}_{0.03}$ | $\mathbf{2.10}_{0.01}$ | $\mathbf{2.22}_{0.05}$ | $\mathbf{2.24}_{0.04}$ | $\mathbf{2.24}_{0.06}$ | $\mathbf{2.19}_{0.03}$ | $\mathbf{2.23}_{0.00}$ | $\mathbf{2.14}_{0.08}$ | $\mathbf{2.45}_{0.11}$ | $\mathbf{2.11}_{0.20}$ |

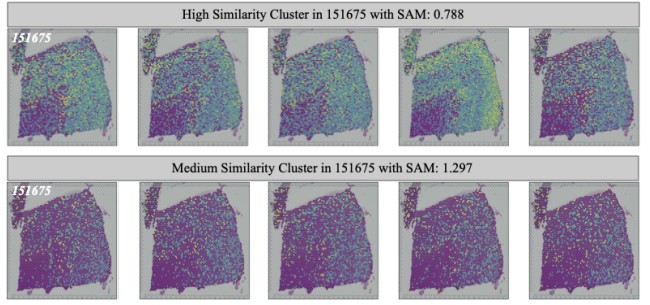
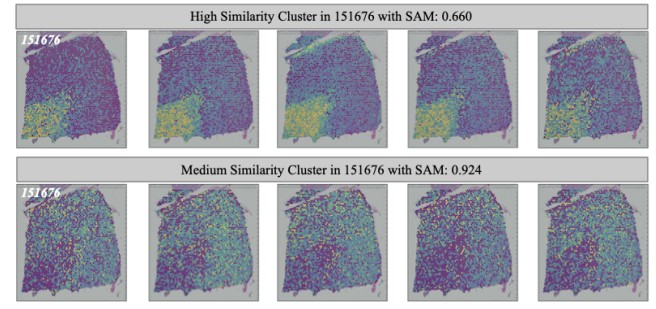

**Figure 3: SGEPs from clusters generated by *DARLC* in dataset (151675, 151676) with high and medium intra-cluster similarity.**

*4.1.3 Cluster Number Inference.* Given the number of image clusters is not known a priori, *DARLC* can estimate this number using a seeding similarity matrix $\mathcal{S} \in \mathbb{R}^{N \times N}$, where $N$ represents the number of images [44]. $\mathcal{S}$ is transformed to a graph Laplacian matrix (refer to Supplementary 1.2), $\mathbb{L} \in \mathbb{R}^{N \times N}$ as follows:

$$\mathcal{S}' = \mathcal{S} + \mathcal{S}^2, \qquad (25)$$

$$\mathbb{L} = \mathbb{D}^{-\frac{1}{2}} \mathcal{S}' \mathbb{D}^{-\frac{1}{2}}. \qquad (26)$$

Here, $\mathbb{D} \in \mathbb{R}^{N \times N}$ represents the degree matrix of $\mathcal{S}$, and $\mathcal{S}'$ aims to enhance the similarity structure. The eigenvalues of $\mathbb{L}$ are then ranked as $\lambda_{(1)} \leq \lambda_{(2)} \leq \cdots \leq \lambda_{(N)}$. The number of clusters, denoted as $K$, is inferred as:

$$K = \text{argmax}_i \left\{ \lambda_{(i)} - \lambda_{(i-1)} \right\}, \quad i = 2, 3, \cdots, N. \qquad (27)$$

**Table 2: Gene-gene interaction prediction results based on the three types of gene image embeddings. The superior method is bolded.**

| Case | ACC (%) | | | | | |
|---|---|---|---|---|---|---|
| | 151671 | 151672 | 151673 | 151674 | 151675 | 151676 |
| iBOT | 66.58 | 68.73 | 68.53 | 65.88 | 68.03 | 67.48 |
| *DARLC*-C1 | 76.18 | 77.20 | 74.71 | 73.64 | 77.85 | 71.50 |
| *DARLC*-full | **78.11** | **78.97** | **77.66** | **78.74** | **78.65** | **73.59** |

*4.1.4 Baselines.* The image clustering benchmark methods include two classic deep clustering method, DEC [41] and DAGMM [49], as well as five SOTA methods that either improve DEC (e.g.,EDESC [3] and IDCEC [29]) or incorporate CL (e.g., CC [25], DCP [27], and CVCL [7]). In addition, we include a benchmark method consisting of iBOT [47], which integrates CL and MIM for representation learning, and a boosted GMM for clustering, denoted as iBOT-C.

*4.1.5 Implementation Details.* The GAT for data augmentation adopts an encoder including a single attention head with C-512-30 network structure, and a symmetric decoder. In *Module I*, the shared encoder structure is a ViT comprising four transformer blocks, each having four attention heads, for processing $75 \times 75$ input images segmented into $4 \times 4$ patches in our case. The MIM decoder follows a D-128-256-512-1024-75*75 residual network. The *Module I* is pre-trained for 50 epochs with a learning rate of 0.001. The nonlinear projection head that bridges *Module I* and *II* is a two-layer MLP for normalizing image representations to a dimension size of 32. The iterative joint optimization of representation learning and clustering continues for 50 epochs using Adam optimizer. Given the absence of ground truth in gene cluster labels, we heuristically determine the number of gene clusters for our experiments to achieve an average cluster size of 30 genes, approximating the typical size of a gene pathway [46].

*4.1.6 Evaluation Metrics.* Without ground truth cluster labels, we evaluate the clustering results using the Davies-Bouldin index (DBI) metric [10]:

$$ \text{DBI} = \frac{1}{K} \sum_{i=1}^{K} \max_{i \neq j} \frac{d_i + d_j}{d_{(i,j)}}, d_i = \frac{1}{|C_i|} \sum_{j=1}^{|C_i|} \delta_{c_i, j}, \tag{28} $$

where $K$ is the number of clusters, $C_i$ the samples in cluster $i$, $\delta_{i,j}$ the distance between instances $i$ and $j$, $c_i$ the centroid of cluster $i$. Cluster width $d_i$ is the mean intra-cluster distance to $c_i$, and $d_{(i,j)} = \delta_{c_i, c_j}$ measures the distance between clusters $i$ and $j$. DBI quantifies the clustering efficiency by measuring the ratio of intra-cluster compactness to inter-cluster separation, with lower scores indicating better clustering. To evaluate clustering from different perspectives, we use two DBI metrics, DBIP and DBIE, based on Pearson and Euclidean distances, respectively [34].

**Table 3: Spatial cofunctional gene clustering results using three image embeddings, with the best method in bolded**

| Case | NMI (%) | | | | | |
|---|---|---|---|---|---|---|
| | 151671 | 151672 | 151673 | 151674 | 151675 | 151676 |
| iBOT | 20.56 | 15.98 | 15.41 | 27.05 | 23.17 | 25.62 |
| *DARLC*-C1 | 66.81 | 67.81 | 70.91 | 65.13 | 74.62 | 75.85 |
| *DARLC*-full | **78.11** | **71.28** | **79.23** | **74.34** | **75.20** | **75.91** |

| Case | ARI (%) | | | | | |
|---|---|---|---|---|---|---|
| | 151671 | 151672 | 151673 | 151674 | 151675 | 151676 |
| iBOT | -2.66 | -5.06 | -5.64 | 4.29 | -0.80 | 2.92 |
| *DARLC*-C1 | 53.41 | 51.61 | 55.58 | 45.88 | 48.97 | 59.83 |
| *DARLC*-full | **59.44** | **53.02** | **65.82** | **36.25** | **59.83** | **54.50** |

## 4.2 Clustering Sparse, Noisy Images of Spatial Gene Expressions

Table 1 showcase the performance of *DARLC*, compared to eight benchmark methods, in clustering gene spatial expression images of 12 real ST datasets, evaluated by the DBIE and DBIP scores, respectively. For each dataset, the experiment is repeated ten times to obtain the mean and standard deviation of each method's scores. *DARLC* consistently achieves the lowest scores in both DBIE and DBIP across all datasets, highlighting its superiority in generating clusters consisting of spatially similar and coherent images. This superiority can be attributed to *DARLC*'s features in integrating MIM and CL, generating more plausible augmented data, and the robust and adaptable clustering algorithm, as substantiated in our ablation study. In contrast, benchmark methods adopt varied strategies: IDCEC and EDESC leverage a convolutional autoencoder for extracting visual features; iBOT+boosted GMM adopts conventional data augmentation; CC, CVCL and DCP rely solely on CL for representation learning; DAGMM employs an outlier-sensitive GMM for clustering. However, these methods generally demonstrate unstable and suboptimal performance. Overall, compared to the best-performing benchmark method, *DARLC* achieves an average reduction of approximately 24.89% in DBIE and 14.39% in DBIP across all datasets. Finally, *DARLC* clusters are divided into high and medium-quality groups using spectral angle mapper (SAM) metric scores [22], with lower SAM indicating greater intra-cluster similarity. To provide a visual illustration of *DARLC*'s clustering performance, we randomly select one cluster from both the high- and medium-quality groups within each of the 151675 and 151676 datasets. From each of the selected clusters, we then randomly select five genes to be displayed in Figure 3. Figure 3 clearly demonstrates that *DARLC* effectively groups images with similar expression patterns into the same cluster.

## 4.3 Evaluating *DARLC*-generated Gene Image Representations

In this section, we present a comprehensive evaluation of gene image representations produced by the fully implemented *DARLC*

model (denoted as *DARLC*-full). First, we assess whether representations generated by *DARLC*-full capture corresponding biosemantics, particularly through pathway enrichment analysis, compared to original gene expressions. Furthermore, we extend our investigation to specific critical downstream tasks: predicting interactions between genes and clustering genes based on spatial cofunctionality. These evaluations are conducted across six distinct datasets (151671-151676). Additionally, gene image embeddings generated by iBOT and a variant of *DARLC* (denoted as *DARLC*-C1), which is deprived of *Module II*, serve as baselines.

*4.3.1 Gene-gene Interaction Prediction.* We employ an MLP-based classifier for predicting gene pair interactions. This is achieved by linear probing using gene image representations generated by *DARLC* and baseline methods (see Supplementary 1.4). We follow the methodology in [15], which is based on the Gene Ontology, to acquire the gene-gene interaction ground truth. Theoretically, gene image representations with richer semantic meanings should yield more accurate predictions. As shown by the accuracy scores in Table 2, the classifier yields the most accurate predictions (77.62%±1.85%) using embeddings generated by *DARLC*-full, and the second most accurate predictions (75.18%±2.17%) using embeddings generated by *DARLC*-C1, followed by the predictions using iBOT-generated embeddings (67.54%±1.03%).

*4.3.2 Spatially Cofunctional Gene Clustering.* Spatially cofunctional genes are those belonging to the same gene family and exhibit similar spatial expression patterns [11]. Their family identities can serve as labels to evaluate the quality of gene image embeddings via clustering. Specifically, our evaluation involves five spatially cofunctional genes from each of the HLA, GABR, RPL, and MT gene families (see Supplementary Figure 1). The Leiden algorithm [36] is used to cluster gene image embeddings generated by *DARLC*-full and the baseline methods. The clustering results, evaluated using the normalized mutual information (NMI) and adjusted rand index (ARI) scores, as shown in Table 3, demonstrate that Leiden yields the most accurate clustering with *DARLC*-full and the second most accurate with *DARLC*-C1.

In summary, these results collectively highlight the effectiveness of the joint clustering (i.e., *DARLC*-full surpasses *DARLC*-C1) and GAT-based data augmentation (i.e., *DARLC*-C1 surpasses iBOT) in enhancing the quality of gene image representations.

## 4.4 Ablation Study

Here, we conduct a series of ablation studies on six ST datasets (151671-151676) to investigate the contributions of *DARLC*'s components in image clustering. The results, detailed in DBIE and DBIP scores, are presented in Table 4. Notably, *DARLC*'s performance declines most with the CL branch removal ("w/o CLR"), followed by the elimination of *Module II* ("w/o SMM") and the substitution of the robust SMM with an outlier-sensitive GMM ("SMM → GMM"). We also observed that employing traditional image smoothing methods such as Gaussian Kernel Smoothing (GKS) instead of GAT ("GAT → GKS") on SNIs leads to a decline in *DARLC*'s performance. Additionally, removing either $\mathcal{L}_{\ell ap}$ ("w/o $\mathcal{L}_{\ell ap}$") or $\mathcal{L}_{size}$ ("w/o $\mathcal{L}_{size}$") from the optimization decreases *DARLC*'s performance, indicated

**Table 4: Ablation study results across six datasets for components in *Module I &II* and regularization terms. The best result is bolded.**

| Case | DBIE↓ | | | | | |
|---|---|---|---|---|---|---|
| | 151671 | 151672 | 151673 | 151674 | 151675 | 151676 |
| w/o CLR | 14.35 | 16.99 | 8.70 | 9.23 | 10.36 | 15.12 |
| GAT → GKS | 9.35 | 10.37 | 8.98 | 9.08 | 8.90 | 8.95 |
| w/o SMM | 11.84 | 12.19 | 11.33 | 11.81 | 11.49 | 12.35 |
| SMM → GMM | 11.49 | 11.36 | 11.16 | 11.31 | 11.05 | 11.37 |
| w/o $\mathcal{L}_{\ell ap}$ | 8.32 | 10.76 | 9.02 | 8.10 | 9.22 | 8.12 |
| w/o $\mathcal{L}_{size}$ | 10.38 | 8.60 | 8.11 | 9.34 | 8.00 | 8.28 |
| w/o Pretraining | 8.22 | 8.74 | 8.38 | 8.18 | 7.87 | 9.09 |
| *DARLC*-full | **7.65** | **8.06** | **7.90** | **7.78** | **7.41** | **8.09** |

| Case | DBIP↓ | | | | | |
|---|---|---|---|---|---|---|
| | 151671 | 151672 | 151673 | 151674 | 151675 | 151676 |
| w/o CLR | 4.21 | 4.21 | 3.51 | 3.97 | 3.82 | 3.69 |
| GAT → GKS | 2.35 | 2.29 | 2.62 | 2.61 | 2.63 | 2.37 |
| w/o SMM | 2.49 | 2.35 | 2.29 | 2.66 | 2.66 | 2.36 |
| SMM → GMM | 2.53 | 2.26 | 2.31 | 2.51 | 2.84 | 2.26 |
| w/o $\mathcal{L}_{\ell ap}$ | 2.33 | 2.27 | 2.26 | 2.31 | 2.66 | 2.17 |
| w/o $\mathcal{L}_{size}$ | 2.29 | 2.22 | 2.25 | 2.25 | **2.31** | 2.19 |
| w/o Pretraining | 2.36 | 2.40 | 2.30 | 2.65 | 2.53 | 2.42 |
| *DARLC*-full | **2.24** | **2.19** | **2.23** | **2.14** | 2.45 | **2.11** |

by higher DBIE and DBIP scores. Lastly, "w/o Pretraining" showcases *DARLC*'s performance at the same clustering training epoch as the complete model but without the initial "warm up" pretraining of *Module I*. The relative underperformance in the "w/o Pretraining" scenario suggests a slower training convergence compared to the complete model.

Overall, all key components of *DARLC* have proven effective, particularly the CL module. This is due to: a) The discontinuous and sparse nature of SNIs, which greatly benefits from the enhanced discriminability provided by CL. b) CL's use of smoothed images as augmented data, which serve as effective priors for guiding *DARLC* to discern the primary visual patterns of SNIs.

## 5 CONCLUSION

In this study, we introduce *DARLC*, a novel algorithm for joint learning of representations and cluster assignments for SNIs. *DARLC* features in its enhanced data augmentation technique, comprehensive and potent representation learning approach that integrates MIM, CL and clustering, as well as robust and adaptable clustering algorithm. These features collectively contribute to *DARLC*'s superiority in both image representation learning and clustering, as evidenced by our extensive benchmarks over multiple real datasets and comprehensive ablation studies.

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
