# OpenReview forum: "Dual Advancement of Representation Learning and Clustering for Sparse and Noisy Images"
_acmmm.org/ACMMM/2024/Conference — MM2024 Poster_

### Official Review · Reviewer_8uUw · 2024-05-24

**Rating:** 5
**Confidence:** 2

**Summary:**

In this paper, the authors propose a clustering framework for sparse and noisy images, which integrating contrastive learning, masked image modeling and deep clustering into a cohesive process for representation learning. Extensive experiments are conducted on 12 real ST datasets to verify the effectiveness of the proposed method.

**Strengths:**

1. The paper is well-written and the structure is well organized.
2. The experiments are numerous and plentiful, and well demonstrate the validity of the methodology.
3. The derivation The proofs of some formulas are detailed and theoretically complete.

**Limitations:**

In this paper, the authors propose a clustering framework for sparse and noisy images, which integrating contrastive learning, masked image modeling and deep clustering into a cohesive process for representation learning. Extensive experiments are conducted on 12 real ST datasets to verify the effectiveness of the proposed method. I have some concerns as follows:
1. As the authors state in their contributions, the authors combine the techniques of contrast learning, self-paced learning, and deep clustering for learning the representations of SNIs, which appears to be an incremental effort. Therefore, what are the authors' motivations for using these techniques, and what particular problems in representation learning of  SNIs can these techniques address?
2. Many components are involved in the proposed method, such as the data generation based on GAT and self-paced learning for clustering etc. Therefore, the time complexity of the algorithm will be high, and the authors should give a detailed time complexity analysis and runtime comparison.
3. The authors learn that good representations of SNIs are used for clustering, and as far as I know, the common evaluation metrics for clustering are ACC, NMI, ARI, etc., and I am more concerned in how the proposed algorithm performs on mainstream metrics versus comparative algorithms in Table I.

**Suitability:**

2

---

### Official Review · Reviewer_H5J9 · 2024-05-25

**Rating:** 4
**Confidence:** 3

**Summary:**

This paper proposes the DARLC framework, which enhances image representation and clustering quality through contrastive learning and a graph attention network, demonstrating superior performance in handling sparse and noisy images.

**Strengths:**

1. The paper proposes an effective method to solve image representation learning and clustering.

2. The paper is well-written.

**Limitations:**

1. The proposed method seems to be a stack of existing techniques, including contrastive learning, graph attention networks, and the Student’s t mixture model. While these techniques each have their strengths, merely stacking them does not necessarily lead to significant innovation. The authors need to clearly articulate the unique technical and theoretical contributions of their approach.

2. The paper does not provide a detailed discussion of the computational complexity of the proposed method. Complexity is a crucial consideration when handling large-scale sparse and noisy images. The authors should include a complexity analysis, covering both time and space complexity, to assess the feasibility and efficiency of their method in practical applications.

3. Although the paper evaluates the method on 12 real-world SNI datasets, the scale and diversity of these datasets might not be sufficient to comprehensively validate the effectiveness of the method. It is recommended to conduct experiments on larger and more diverse datasets to better demonstrate the robustness and general applicability of the method.

**Suitability:**

2

---

### Official Review · Reviewer_FyRu · 2024-05-31

**Rating:** 4
**Confidence:** 2

**Summary:**

Sparse and noisy images (SNIs) pose significant challenges to effective representation learning and clustering, which are essential for thorough data analysis and interpretation.
This paper proposed the Dual Advancement of Representation Learning and Clustering (DARLC), an innovative framework that leverages contrastive learning to enhance the representations derived from masked image modeling.
The proposed framework offers a comprehensive approach that improves the learning of representations by enhancing their local perceptibility, distinctiveness, and understanding of relational semantics.
Experiment results demonstrate DARLC’s superiority over current state-of-the-art methods in both image clustering and generating representations that accurately reflect biosemantics content and gene interactions.

**Strengths:**

1. The authors have done well in the demonstration of the studied problem and the overall framework. Such as the Figure 1.

2. I agree with the authors that Sparse and noisy images (SNIs) are interesting and important to work on.

3. The experiment evaluations are comprehensive and reasonable, the authors evaluated their method on 12 real-world datasets and compared their method with multiple baselines.

3. Although I can't fully examine the correctness of the proof in the Appendix. I think the authors have done well in deriving of the gradients, which shows their certain abilities in statistics and optimization.

**Limitations:**

1. The first apparent weakness is the quality of the writing of the abstract. Note that I think this is relatively minor and easy to fix, but the grammar and vocabulary should be improved. For example "demonstrat" should be "demonstrate".

2. I believe the authors should provide a detailed explanation of the reasons for choosing techniques such as CL, MIM, and SMM. It will be good to explain in detail why these techniques are more effective for SNIs. The current demonstration is more like a combination of techniques or a technical report rather than a unified paper. One example is in section 3.2.2 "specifically designed for generating image patch embeddings in the context of SNIs." I think a brief explanation for why this "adapted version of MAE" is more effective for SNIs is required.

3. The authors employ a lot of symbols, I think a summary of all used symbols will help readers to understand this work better.

4. In section 3.3.2, there is a reference error, "Here, L𝑟𝑒𝑐 and L𝑐𝑙𝑟 remain the same as in Equations 7 and ?? except being calculated on the batch-level."

5. I suggest authors add a summary sentence before each section, especially the methodology section, which can help readers figure out the structure of this section.

6. I think a more detailed introduction of the used datasets will be useful.

7. There are multiple reference errors in the Appendix 1.4. For example, "which is a basic MLP-based network as described in Du et al., 2019 [?]". I suggest authors inspect all these errors carefully.

Overall, please understand that I agree with the authors that Sparse and noisy images (SNIs) are important and difficult to work on. However, I think authors should put more effort into the clarity of the writing, especially in the methodology section.

**Suitability:**

3

---

### Official Review · Reviewer_ujdF · 2024-06-04

**Rating:** 4
**Confidence:** 3

**Summary:**

The paper introduces DARLC, a novel framework for joint representation learning and clustering of sparse and noisy images (SNIs). DARLC integrates contrastive learning (CL), masked image modeling, and deep clustering into a cohesive process, enhancing image clustering performance and benefiting semantic distance-based tasks. It addresses the challenges posed by SNIs, such as uninformative regions and high noise levels, by introducing a GAT-based data augmentation method to generate smoothed images that improve contrastive learning. Furthermore, DARLC employs a Student’s t mixture model for robust and adaptable soft clustering, which down-weights extreme values and adjusts degrees of freedom to suit SNIs. The framework also addresses the class collision problem in CL by using the clustering loss to regularize it.

**Strengths:**

- This paper proposes a novel algorithm for joint learning of representations and cluster assignments for SNIs and verifies it on spatial transcriptome data.
- In the experiment, the method proposed in the paper significantly outperforms the baseline methods.
- The paper is well-written.

**Limitations:**

### Weakness:
- The DARLC proposed in this paper uses the t distribution for subsequent analysis. There is nothing wrong with the technical process, but there seems to be no good explanation and description of why the t distribution was chosen. In other words, can other tractable distributions and corresponding mixture models be chosen? The answer may be yes, in which case the author needs to explain the rationality and advantages of the t distribution modeling.
- Please explain the intuition behind Eq(25) and Eq(27), and why use the degree matrix of $S$ to normalize $S'$ in Eq(26)? What does the distribution of $\lambda_{i} - \lambda_{i-1} $ look like?
- How do you choose $k$ in line 284?

### Minor points:
- Extra '.' in the caption of section 3.2.

**Suitability:**

2

---

### Meta-Review · Area_Chair_FSwy · 2024-07-02

**Recommendation:** Accept (Poster)
**Confidence:** 5

**Metareview:**

All reviewers agree that this paper has its own contribution to the field, although there are some issues in writing and detailed representation. We recommend the authors taking all comments from reviewers to further improve this paper.